# Alternative splicing and residual function potentially expand the therapeutic landscape of the CFTRdele2ins182 variant

Cristina Pastorino[1], Ludovica Menta[2], Emanuela Pesce[1,2], Mariateresa Lena[2], Valeria Tomati[1], Valeria Capurro[1], Marco Di Duca[1], Vito Terlizzi[3], Andrea Gramegna[4,5], Francesco Blasi [ID][4,5], Carlo Castellani[6], Serena Cappato[1], Federico Zara[1,2], Nicoletta Pedemonte [ID][1]*, Renata Bocciardi[1,2]

**1** Medical Genetics Unit, IRCCS Istituto Giannina Gaslini, Genoa, Italy, **2** Department of Neurosciences, rehabilitation, Ophthalmology; Genetics and Maternal and Child Sciences (DINOGMI), University of Genoa, Genoa, Italy, **3** Department of Pediatric Medicine, Meyer Children's Hospital IRCCS, Cystic Fibrosis Regional Reference Centre, Florence, Italy, **4** Respiratory Unit and Cystic Fibrosis Adult Center, Fondazione IRCCS Ca' Granda Ospedale Maggiore Policlinico, Milan, Italy, **5** Department of Pathophysiology and Transplantation, University of Milan, Milan, Italy, **6** Cystic Fibrosis Center, IRCCS Istituto Giannina Gaslini, Genoa, Italy

* nicolettapedemonte@gaslini.org

## Abstract

This study investigates the molecular and functional consequences of rare *CFTR* variants, particularly focusing on the complex allele [186-13C > G; 1898 + 3A > G] and the CFTRdele2ins182 rearrangement. Using patient-derived nasal epithelial cells, the research characterized the transcripts produced by these variants, revealing that CFTRdele2ins182, previously considered a null allele, generates alternative mRNA isoforms, one of which potentially encodes a partially functional CFTR protein. Functional assays in both heterologous and patient-derived cell models explored the impact of CFTR modulators on these variant proteins. While some rescue of CFTR activity was observed with specific modulator combinations in certain variants, the study highlights the complexity of genotype-phenotype correlations in CF and emphasizes the importance of personalized functional characterization of rare *CFTR* variants to guide therapeutic strategies. The findings suggest that even variants thought to be null alleles may produce proteins with residual function, opening avenues for developing targeted therapies for a broader range of CF patients.

## Introduction

Cystic Fibrosis (CF) is a severe genetic disease caused by variants in the Cystic Fibrosis Transmembrane Conductance Regulator (*CFTR*) gene that encodes for the CFTR protein, a chloride channel localized in the apical membrane of the epithelial cells [1–3]. CF has an incidence between 1/3000 and 1/6000 in the Caucasian populations [4] and more than 2100 variants of the *CFTR* gene have been reported,

**Data availability statement:** All relevant data are within the manuscript and its Supporting Information files.

**Funding:** Fondazione per la Ricerca sulla Fibrosi Cistica grants n. FFC #3/2023, n.FFC #9/2019 and n. FFC #10/2021; Italian Ministry of Health [grants n. GR-2018–12367126 and PNRR-MR1-2023-12378412] and through Cinque per mille (5Xmille 2019 grant 5M-2019-23680413) and Ricerca Corrente. The funders had no role in study design, data collection and analysis, decision to publish, or preparation of the manuscript.

**Competing interests:** The authors have declared that no competing interests exist.

however not all of them exert pathogenetic effects. To date, 1085 variants have been well defined as CF-causing (CFTR2 database accessed on January 15th, 2025). Mutations of the *CFTR* gene are grouped into functional classes, according to their molecular and functional consequences [5]. CFTR class I mutations cause absence of *CFTR* mRNA or protein as in the case of variants affecting the acceptor or donor splicing sites or nonsense or frameshift mutations that introduce premature termination codons (PTCs) [6]. Class II variants generate a protein that fails to traffic to the cell surface due to misfolding and premature degradation by the proteasome [6]. The deletion of phenylalanine at position 508 (F508del), the most frequent variant in CF patients worldwide, belongs to this class of mutants [6]. Defective channel gating, with a reduced open channel probability, and reduced conductance are the peculiar features of class III and IV mutations, respectively (De Boeck and Amaral, 2016). Class V includes mutations that alter the splicing, reducing the levels of normal CFTR protein [6]. Finally, class VI groups the mutations that impair the stability of CFTR protein on the plasma membrane, leading to its internalization and degradation [6].

The classification of CF variants, however, is not always straightforward since many mutations cause mixed functional defects and may be included in more than one class [7]. One of the most representative examples is the case of F508del, that, besides the misfolding defect, exhibits also defective gating and reduced stability on the plasma membrane [7]. For this reason, more recently, it was proposed to classify CF-causing variants based on their response to CFTR pharmacological modulators rather than on their impact at the protein level [8].

Since 2012, several CFTR-modulating drugs have been developed and are now used in clinical practice for people with CF (pwCF) carrying specific mutations. These drugs are small molecules able to bind to CFTR protein and partially rescue specific defects [2]. Among them, potentiators are compounds that improve the gating defect typical of class III mutations [2]. To date, Ivacaftor (IVA) is the only approved potentiator for CF patients carrying a group of specific gating mutations [2]. Correctors are, instead, compounds able to rescue the folding and trafficking defect of class II mutations [2]. Correctors include Lumacaftor (LUM) [9], Tezacaftor (TEZ) [10] and Elexacaftor (ELX) [11]. When correctors are combined, as in the ELX/TEZ combination, they exert additive effects, thus improving mutant CFTR rescue [12,13]. Furthermore, the combination of correctors and potentiators has been developed to target mutations that cause different CFTR defects [2,7].

While modulators therapy has revolutionized the care of CF patients, to date, a significant fraction of pwCF (in Italy, 30% approximately) carry variants that are ineligible to treatment with these highly effective modulators [14]. Among these variants, there are missense mutations, insertions, deletions, frameshifts as well as nonsense and splicing mutations.

Splicing variants account for approximately 12% of all reported CFTR variants [15]. In the Cystic Fibrosis Mutation Database (www.genet.sickkids.on.ca) 231 splicing variants are reported, counting for 10.9% of all the mutations listed in this database. Most of the splicing variants affect the canonical AG (3' splicing site) and GU (5' splicing site) motifs; however, it is estimated that 1–2% of *CFTR* variants are deep intronic

splicing mutations that may be elusive [15]. From this point of view, the routine use of next generation sequencing (NGS) technologies will contribute to improve the detection rate of such variants.

Determining the consequences of a sequence variation at the transcript and/or protein level is not always straightforward. The development of new predictive models based on the use of patient-derived intestinal or nasal epithelia (HNE) cells [16–18] has allowed to perform both targeted molecular analyses of the *CFTR* transcript and functional studies of the CFTR protein in a native system, to unmask the defect caused by a single variant. Furthermore, through this system, it is possible to get insights into the responsiveness of patients' variants to CFTR-modulators. This approach has a relevant value as already confirmed in several recent studies [16–22].

In the frame of a project aiming at characterizing rare variants still orphan of therapy, we recruited an individual compound heterozygous for a complex allele [186-13C > G; 1898 + 3A > G] and the variant CFTRdele2ins182. Both the 1898 + 3A > G and CFTRdele2ins182 variants are known CF-causing alleles [23–25], whereas the additional 186-13C > G substitution has never been characterized from a molecular or functional point of view, and it is reported with a very low frequency in population databases.

We provide here a functional and molecular characterization of the CFTRdele2ins182 rearrangement and the complex allele [186-13C > G; 1898 + 3A > G] using *ex vivo* HNE cells derived from the recruited compound heterozygous patient, carrying a genotype that *a priori* is not eligible for the treatment with CFTR modulators. Interestingly, we found that the CFTRdele2ins182, considered as a null-allele, may be transcribed in alternative forms of *CFTR* mRNA, of which one may lead to the synthesis of a mutant CFTR protein retaining some activity.

## 2. Materials and methods

### 2.1. Patients under study

A CF individual (donor ID: FI077) compound heterozygous for the CFTR2ins182 variant and a complex allele [186-13C > G;1898 + 3A > G] was initially included in our studies. Three additional patients compound heterozygous for the CFTR2ins182 variant and the R1066H (donor ID: GE202), or the 1717-1G > A (donor ID: FI066), or the c.870–1110_1113delTAAG (donor ID: MI253) variant were recruited to corroborate our findings on the CFTR2ins182 variant. HNE cells derived from two healthy subjects (donor IDs: Ctr032 and Ctr070) were included in our studies as controls. Conversion between Legacy names and *HGVS* nomenclature is reported in S1 Table.

### 2.2. Primary nasal epithelial cell culture

Primary human nasal epithelial cells (HNEC), collected by nasal brushing of both nostrils, were isolated and cultured as previously described [20,22]. To obtain differentiated epithelia, HNEC were seeded at high density on Snapwell permeable support (code 3801, Corning, Tewksbury, MA, United States) using PneumaCult-Ex Plus Basal Medium (Stemcell Technologies, Vancouver, Canada) as proliferative medium on both the apical and the basolateral side. After 24 hours, the medium was removed and replaced (on the basolateral side only) with PneumaCult-ALI Basal Medium (Stemcell Technologies, Vancouver, Canada) to promote epithelial differentiation under air-liquid interface condition. The differentiative medium was changed every other day for about 16–18 days to achieve epithelial differentiation. Nasal epithelia with transepithelial resistance (Rt) ranging between 400 and 600 $\Omega$ cm$^2$ (thus forming electrically tight epithelia [17,22]) were used for the short-circuit current analysis.

### 2.3. Short-circuit current recordings

Snapwell supports with the differentiated nasal epithelia were mounted in a vertical diffusion Ussing Chamber with internal fluid circulation, as previously described [20,22,26]. Both apical and basolateral hemichambers were filled with 5 mL of a solution containing (in mM) 126 NaCl, 0.38 $KH_2PO_4$, 2.13 $K_2HPO_4$, 1 $MgSO_4$, 1 $CaCl_2$, 24 $NaHCO_3$, and 10 glucose. Both

sides were continuously bubbled with a gas mixture containing 5% $CO_2$–95% air and the temperature of the solution was kept at 37 °C. The transepithelial voltage was short-circuited with a voltage-clamp (DVC-1000, World Precision Instruments, Sarasota, FL, USA; VCC MC8 Physiologic Instruments, Reno, NV, USA) and connected to apical and basolateral chambers with Ag/AgCl electrodes (EK1, WPI Instruments; two for the voltage and two for current) and with agar bridges (1 M potassium chloride in 2% agar). The short-circuit current was recorded with a PowerLab 4/25 (AD Instruments, Colorado Springs, CO, United States) analogue-to-digital connected converter to a personal computer.

### 2.4. Chemicals and vectors

The CFTR modulators Ivacaftor and Tezacaftor were purchased from TargetMol (catalog ID:T2588 and T2263, respectively; Wellesley Hills, MA, USA), while Elexacaftor from MedChemExpress (catalog ID: HY-111772; Monmouth Junction, NJ, USA). The final working concentration used for the CFTR modulators were as follows: Elexacaftor, 3 µM; Tezacaftor, 10 µM; ivacaftor, 1 µM (when applied acutely during short-circuit current measurements or for the YFP assay) or 5 µM (for 24 h treatment in the ETI combination). Vectors encoding wildtype-, Dele13-, Dele2,3-; Dele2-, and F508del-CFTR proteins were purchased from Vector-Builder (vector IDs available upon request; Neu-Isenburg, Germany).

### 2.5. Cell culture

Immortalized human bronchial epithelial cells (CFBE41o-) stably expressing the halide-sensitive yellow fluorescent protein (HS-YFP-H148Q/I152L; [27]) were grown in MEM medium (Euroclone) supplemented with 10% FBS, 2 mM L-glutamine, 100 U/mL penicillin, and 100 µg/mL streptomycin (Euroclone).

### 2.6. Transient transfection of CFBE41o- cell line

For the minigene study, CFBE41o-cells were seeded onto 6-well plates and transiently transfected by reverse transfection with 2 µg/well of the mutated, or wild-type, or empty constructs. For the YFP assay, CFBE41o- cells were reverse-transfected onto 96-well plates with 0,2 µg/well of the vectors reported (indicated in the "Chemicals and Vectors" Methods Section).

Lipofectamine 2000 (ThermoFisher Scientific, Waltham, MA, USA) was used as transfection agent in Opti-MEM™ Reduced Serum Medium (ThermoFisher Scientific, Waltham, MA, USA). After 6 h, DNA-lipofectamine complexes were removed and replaced with culture medium without antibiotics. Twenty-four hours after transfection, cells were harvested and processed as described in the "Minigene study" Methods Section or, for the YFP-based assay, were treated with correctors or with vehicle alone (DMSO) at the reported concentrations for an additional 24 hours prior to proceeding with the functional assay.

### 2.7. YFP-based assay for CFTR activity

CFTR activity was determined by the HS-YFP microfluorimetric assay, described in detail in previous studies [13,26,28]. Briefly, prior to the assay, cells were washed with PBS (137 mM NaCl, 2.7 mM KCl, 8.1 mM $Na_2HPO_4$, 1.5 mM $KH_2PO_4$, 1 mM $CaCl_2$, and 0.5 mM $MgCl_2$) and then incubated for 25 min with 60 µL of PBS plus forskolin (20 µM) and Ivacaftor (1 µM) at 37°C, to maximally stimulate the CFTR channel. Cells were then transferred to a microplate reader (FluoStar Galaxy or Fluostar Optima; BMG Labtech, Offenburg, Germany), equipped with high-quality excitation (HQ500/20X: 500 ± 10 nm) and emission (HQ535/30M: 535 ± 15 nm) filters for YFP (Chroma Technology, Bellows Falls, VT, USA). The assay allows YFP fluorescence to be recorded for 14 seconds: 2 seconds before and 12 seconds after the injection of 165 µL of an iodide-containing solution (PBS with Cl- replaced by I-; final I- concentration 100 mM). Data were normalized to the initial fluorescence. To determine the I- influx rate, an exponential function was used to fit the fluorescence data recorded in the final 11 s for each well, and to extrapolate the initial slope (dF/dt).

## 2.8. RNA extraction and cDNA analysis

After Ussing chamber analysis, differentiated epithelia treated with the vehicle alone (DMSO) were collected by scraping in 200 μL microliter of TRIzol reagent (Ambion-ThermoFisher Scientific, Waltham, MA, USA). Total RNA extraction was then performed with the ReliaPrep™ RNA Cell Miniprep Systems kit (Promega Madison, WI, USA) as suggested by the manufacturer. The obtained RNA was checked and quantified with the Nanodrop Spectrophotometer (ThermoFisher Scientific, Waltham, MA, USA).

First strand cDNA was synthesized with the PrimeScript™ High Fidelity RT-PCR Kit (Takara Bio Inc) from 2 μg of total RNA. Qualitative analysis of the CFTR transcript was performed by PCR amplification with specific oligonucleotides (sequences available upon request) spanning the cDNA region bearing the variants under study (CFTRdele2ins182: NM_000492.4:c.54–5811_164+2186del8108ins182; 186-13C>G: rs397508749, NM_000492.4:c.54-13C>G; 1898+3A->G: rs213950, NM_000492.4:c.1766+3A>G; nomenclature according to the *HGVS* guidelines). Briefly, 4 microL of cDNA diluted 1:2 was used as a template to amplify *CFTR* mRNA with the GoTaq G2 Polymerase Mastermix (Promega Madison, WI, USA).

The obtained PCR products were checked by agarose gel electrophoresis, purified by enzymatic digestion with Exo/SAP-IT (ThermoFisher Scientific, Waltham, MA, USA) and then used for direct sequencing. Reactions were set up with a Big-Dye-Terminator v1.1 Cycle Sequencing kit (Applied Biosystems-ThermoFisher Scientific, Waltham, MA, USA) as suggested by the provided protocol, run on a 3130xlGenetic Analyzer (Applied Biosytems-ThermoFisher Scientific, Waltham, MA, USA),and analyzed by the Sequencer 4.7 software (Genecodes).

## 2.9. Minigene study

A 223 bp genomic fragment spanning *CFTR* exon 2 and flanking intronic sequences was obtained by amplification from genomic DNA of a control individual. The obtained PCR products were subcloned with the TOPO™ TA Cloning™ Kit (ThermoFisher Scientific, Waltham, MA, USA) and sequence checked. The186-13C>G variant was introduced by site-directed mutagenesis by using the QuikChange Site-Direct Mutagenesis Kit (Agilent Technologies, Santa Clara, CA, USA) according to the suggested protocol, using specific oligonucleotides (sequences available upon request).

The wild-type and the mutated fragment were then retrieved by EcoRI restriction and transferred into the same site of the pSPL3 splice vector (Tompson SW and Young TL, 2017). The mutated, wild-type and empty constructs were transiently transfected in CFBE 41 o-cells using Lipofectamine™ 2000 (ThermoFisher Scientific, Waltham, MA, USA) as described in "Transient Transfection of CFBE41o- Cell Line" Methods Section. Twenty-four hours after transfection, cells were harvested by scraping and total RNA extracted with the RNeasy Mini Kit (Qiagen, Germantown, MD, USA). After first strand cDNA synthesis, RT-PCR was performed with oligonucleotides specific for the splice vector exons. Finally, the obtained PCR products were visualized on agarose gel and verified by the direct Sanger sequencing.

## 2.10. *In silico* analysis of splicing variants

The spliceAI online tool (https://spliceailookup.broadinstitute.org/) was applied to evaluate the effect of the186-13C>G intronic variant on the splicing of the corresponding mRNA. The software returns a "delta score" that ranges from 0 to 1 corresponding to the probability that the variant affects the splicing at any position within a window around it (+/- 50 bp) with different threshold values [29].

## 2.11. Statistics

The hypothesis of a normal distribution of the data was tested using the Kolmogorov-Smirnov test. Analysis of variance (ANOVA) followed by a post-hoc test was used to avoid the "multiple comparison error". Normally distributed quantitative variables were evaluated by parametric ANOVA. The statistical significance of the effect of individual pharmacological

agent treatments on nasal cell activity was assessed by parametric ANOVA followed by Dunnet's multiple comparison test (all groups versus control group) as a post-hoc test. The ANOVA test followed by the Tukey test (for multiple comparisons) as a post-hoc test was employed in the case of drug combinations. Selected treatment pairs were compared, and statistical significance was tested by ANOVA followed by Bonferroni as a post-hoc test. Normally distributed data are described as mean ± standard deviation (SD), and significances are two-sided. When $p < 0.05$, differences were considered statistically significant.

## 2.12. Ethics statement

The research involving human participants received ethical approval from the Ethics Committee of the IRCCS Istituto Giannina Gaslini (approval number CER 28/2020, February 4th, 2020). All procedures adhered to institutional protocols and applicable national regulations. Written informed consent was obtained from participants or, in case of minors, from participants' legal guardians or next of kin prior to inclusion in the study. The recruitment period started on February 20th, 2020, and ended on December 20th, 2024.

## Results and discussion

In the frame of a wide, multicentric project aiming at charactering rare *CFTR* variants and defining their response to approved CFTR modulators and novel drugs (a process named theratyping, i.e., defining to which type of therapy a specific *CFTR* variant may be responsive), our laboratory recruited more than 400 pwCF that underwent nasal brushing to collect nasal epithelial cells.

A collaborating clinical CF center brought to our attention a CF individual (donor ID: FI077), with a severe medical condition, who is compound heterozygous for the CFTRdele2ins182 variant and the novel complex allele [186-13C > G; 1898 + 3A > G].

The CFTRdele2ins182, is a well-known CF-causing variant consisting of a complex rearrangement including a deletion of approx. 8kb comprising the whole exon 2 and adjacent intronic sequences, and a 182 bp fragment, originating from the duplication of a sequence present in intron 3, inserted into a reversed orientation at the breakpoint of the deletion [24]. Although the absence of exon 2 would be an expected consequence of such an alteration, it was shown that, at the mRNA level, this rearrangement causes, in fact, the skipping of both exons 2 and 3, with the loss of the reading frame and formation of an early termination codon [24]. Therefore, CFTRdele2ins182 is thought to behave as a null allele.

The first variant of the complex allele is the 1898 + 3A > G substitution, which maps in the *CFTR* sequence immediately adjacent to the *CFTR* exon 13 donor splice site, is a known CF-causing variant [23,25]. Minigene assay studies have demonstrated that also in wild-type *CFTR*, exon 13 may be spliced out from a fraction of transcripts, due to the presence of an Intronic Silencer Sequence (ISS) in the adjacent region. The presence of the 1898 + 3A > G variant completely abolishes the inclusion of the exon in mature RNA [30,31]. These functional data have never been confirmed in primary cells from patients.

The additional substitution 186-13C > G, *in cis* with this splicing variant, maps in the intronic sequence adjacent to the splice acceptor of exon 2. It has never been characterized from a molecular point of view, and it is interpreted as variant of unknown significance with a very low frequency in the gnomAD population databases (https://gnomad.broadinstitute.org/).

Based on the genotype, the patient was not considered eligible for treatment with CFTR modulators. Given the presence of a previously unreported, therefore yet uncharacterized, complex allele, we aimed to investigate its consequences at the molecular and functional level.

Patient's nasal epithelial cells (HNEC) were collected by nasal brushing, cultured using established protocols and then differentiated on porous supports into nasal epithelia (HNE) under air-liquid conditions (ALI) [20–22].

We first performed a detailed molecular analysis of the *CFTR* cDNA obtained by reverse transcription of the total RNA obtained from differentiated epithelia.

Sequencing of the RT-PCR products spanning the CFTRdele2ins182 and 186-13C>G variants from exon 1–5 from HNE cells of the patient showed the presence of three different *CFTR* isoforms. As shown in Fig 1A,B, these three transcripts correspond to an mRNA including all exons (a), and two different transcripts with the skipping of exon 2 alone (b) or the concomitant deletion of exon 2 and 3 (c), respectively. This latter causes the formation of a premature stop codon (PTC) and is probably more prone to degradation as suggested by the observation that it is also less represented (lowest peaks at the Sanger sequencing analysis). We also performed the analysis of the cDNA spanning the exons 11–14 (Fig 2C,D) and confirmed the presence of a transcript lacking exon 13 (e) due to the 1898+3A>G substitution, as already described in heterologous systems by minigene assay [30].

The CFTRdele2ins182 has been described as a complex genomic rearrangement that induces the concomitant skipping of both *CFTR* exons 2 and 3 [24]. For this reason, we investigated whether the 186-13C>G variant could affect the *CFTR* splicing, thus leading to the generation of the observed transcript skipping the exon 2 alone. This substitution does not affect the canonical acceptor splicing site of the exon but could still alter the sequence environment required for a proper mRNA processing in this region, at least in a proportion of transcripts. To this purpose we applied the minigene approach.

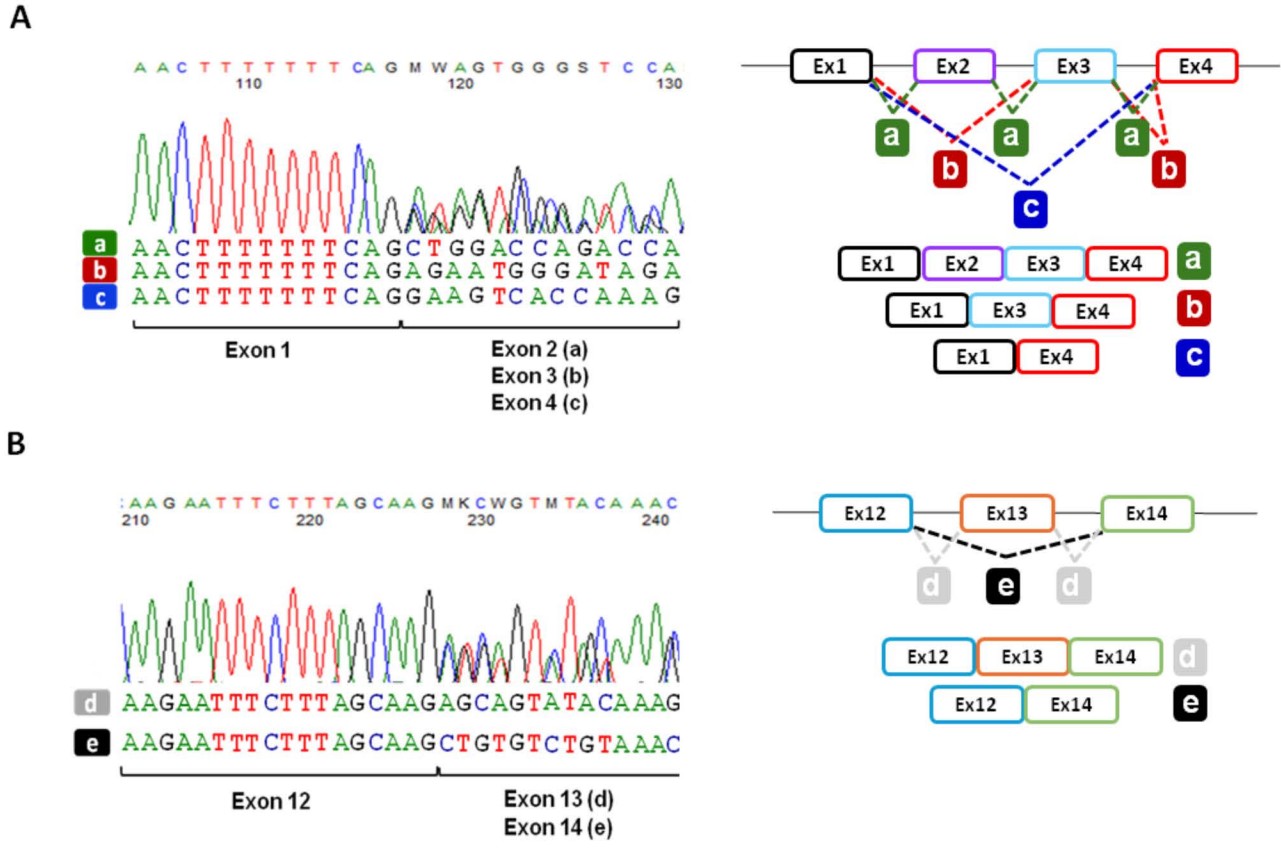

**Fig 1. Molecular analysis of *CFTR* transcripts in FI077 patient-derived HNE cells. A**. Chromatograms obtained by direct sequencing of the RT-PCR products derived from patient FI077 HNEC spanning *CFTR* exons 1 to 5 showing the presence of a transcript including all exons **(a)**, a transcript with the skipping of exon 2 **(b)**, and a transcript with the skipping of exons 2 and 3 **(c)**. The right panel shows a schematic representation of the transcript's composition. **B**. Chromatograms obtained by direct sequencing of the RT-PCR products from patient FI077 HNEC spanning *CFTR* exons 11 to 14 showing the presence of a transcript including all exons (d) and a transcript with the skipping of exon 13 **(e)**. Schematic representation of the observed *CFTR* splicing shown in **B** is provided in the right panel.

The genomic region spanning the 186-13C>G substitution (in wild-type and mutant form) was subcloned into a pSPL3 expression vector to characterize splicing events after transient transfection into a suitable cellular recipient (schematically represented in Fig 2A). In this case we used CFBE41o- cells, a widely used model for studies on the pharmacology and biology of CFTR. As shown in the Fig 2B, the 186-13C>G variant does not appear to alter the splicing of the *CFTR* exon 2 and this was subsequently confirmed by direct sequencing of the obtained RT-PCR products (Fig 2C).

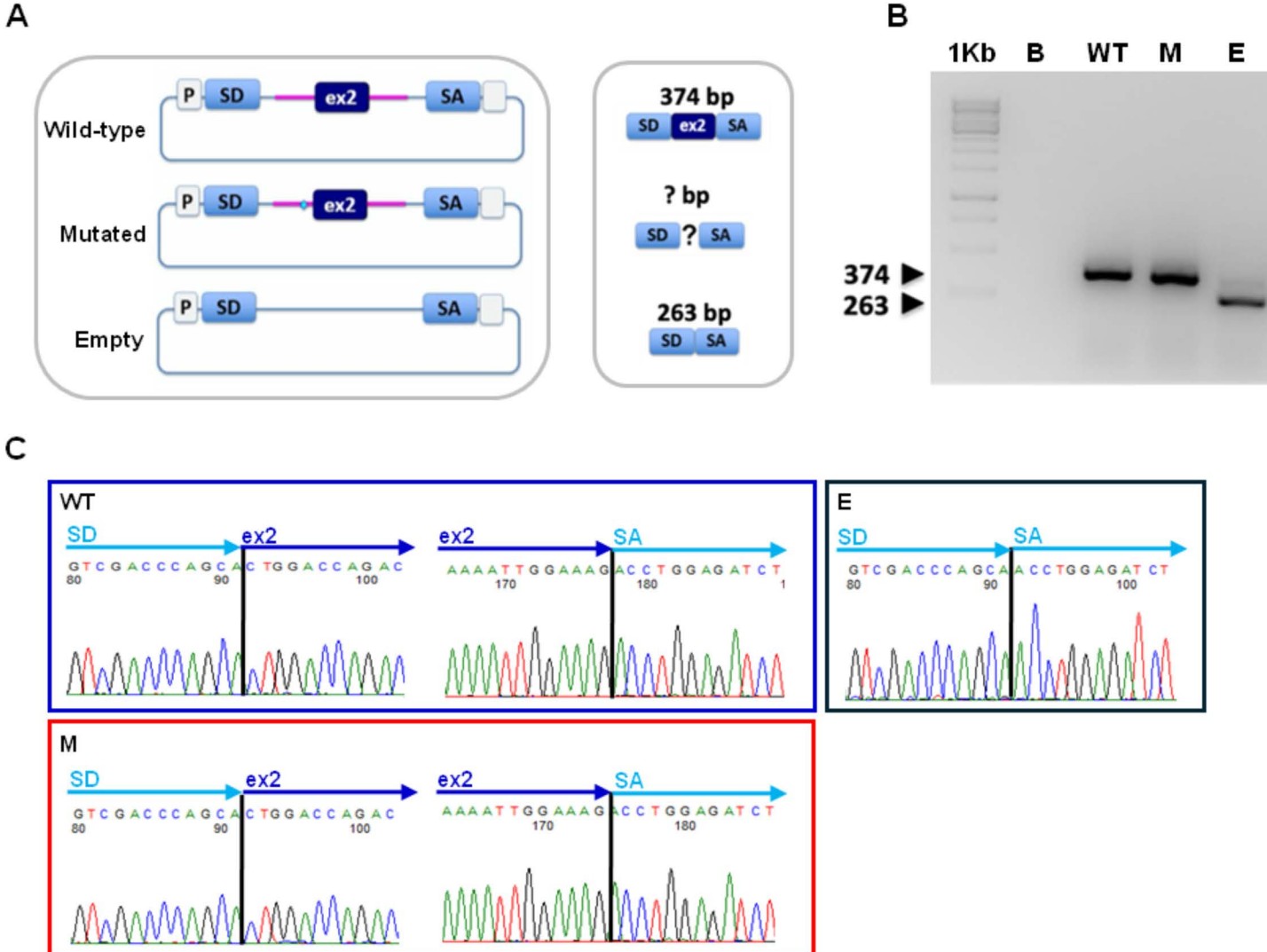

**Fig 2. Minigene approach for the functional characterization of the 186-13C>G variant. A**. Schematic representation of the constructs in pSPL3 expression vector. Both the wild-type and the 186-13C>G carrying genomic fragments spanning the *CFTR* exon 2 and flanking intronic sequences were subcloned into the pSPL3 splicing vector. Expected RT-PCR products generated with oligonucleotides specific for the SD and SA exons of the pSPL3 vector after transient transfection of the wild-type and mutated constructs in CFBE41o- cells together with empty vector as a control are represented in the middle panel. **B**. Agarose gel electrophoresis showing the obtained RT-PCR products: a 374 bp band corresponding to the canonical splicing of *CFTR* exon 2 with the SD and SA flanking exons in CFBE41o- cells transfected with the wild-type (WT) and mutated (M) constructs and as expected, a 263 bp product in cells expressing the empty vector **(E)**. **C**. Chromatograms obtained by direct sequencing of the RT-PCR products showing the junctions between exons. 1Kb, molecular weight marker; B, RT-PCR blank; WT and M, RT-PCR products obtained by the splicing of the pSPL3 wild-type and mutated constructs, respectively; E, RT-PCR product in cells transfected with pSPL3 empty vector.

We thus focused in more depth on the CFTRdele2ins182 rearrangement to verify the composition of the transcripts derived from this allele. To do this, we amplified the *CFTR* cDNA of patient FI077 with a couple of specific oligonucleotides in which the reverse primer, mapping in exon 13, allowed the analysis of the transcripts exclusively derived from the CFTRdele2ins182 allele. As shown in Fig 3A, Sanger sequencing of the obtained RT-PCR products demonstrated the presence of two different mRNAs, one lacking exon 2 only and one without exon 2 and 3, as already reported [24]. As a complementary control, transcripts from the second allele were isolated using specific forward and reverse oligonucleotides designed on exons 2 and 14, thus confirming the presence of the exon 2 and the skipping of exon 13, due to the 1898 + 3 A > G substitution, as expected (Fig 3B).

While the molecular characterization of patient FI077 was in progress, three additional CF patients (donor IDs: FI066, GE202 and MI253) carrying the same complex genomic rearrangement CFTRdele2ins182 were recruited. The first one was compound heterozygous for the CFTRdele2ins182 and 1717-1G > A variants (donor ID: FI066); the second carrying the CFTRdele2ins182/R1066H genotype (donor ID: GE202); the third compound heterozygous for the CFTRdele2ins182 and the c.870–1113_870–1110del variants. The amplification and the sequencing of the RT-PCR products spanning the CFTRdele2ins182 showed, also in these patients, the same pattern of transcripts, with mRNAs lacking exon 2 and both exon 2 and 3 (S1 Fig).

Interestingly, considering the two altered *CFTR* transcripts expressed in the nasal epithelia of the CFTRdele2ins182 carrying patients, the one lacking exon 2 only maintains the correct reading frame, potentially leading to the synthesis of

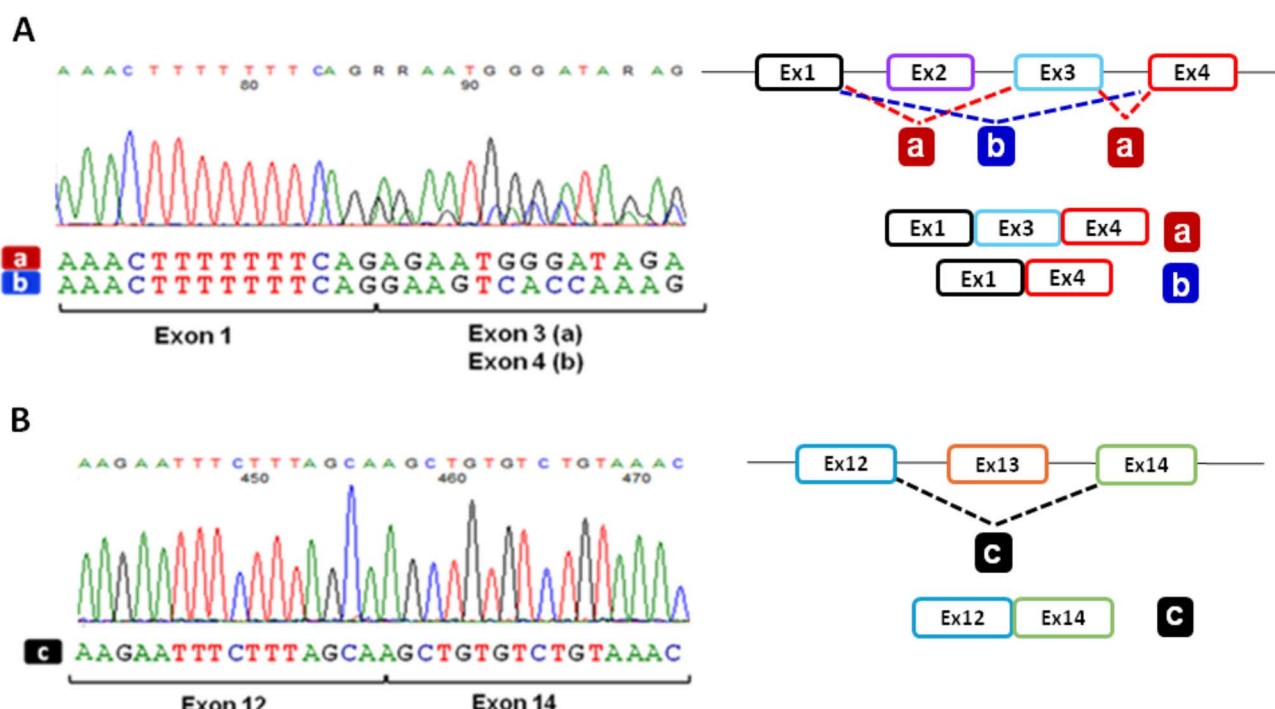

**Fig 3. Analysis of the *CFTR* allele-specific transcripts derived from FI077 patient-derived HNE cells. A**. Chromatograms obtained by direct sequencing of the RT-PCR products spanning exons 1 to 13 with specific reverse oligonucleotide on *CFTR* exon 13 to specifically target the transcript generated from the CFTRdele2ins182 allele. As shown by electropherograms and schematically represented in the right panel, two different transcripts with the skipping of exon 2 (a) and with the concomitant skipping of exons 2 and 3 (b) are detectable. **B**. Chromatograms obtained by direct sequencing of the RT-PCR products obtained with specific forward oligonucleotide mapping in exon 2 showing the presence of a transcript with the skipping of exon 13 **(c)**, the right panel shows the corresponding schematic representation of the observed *CFTR* splicing.

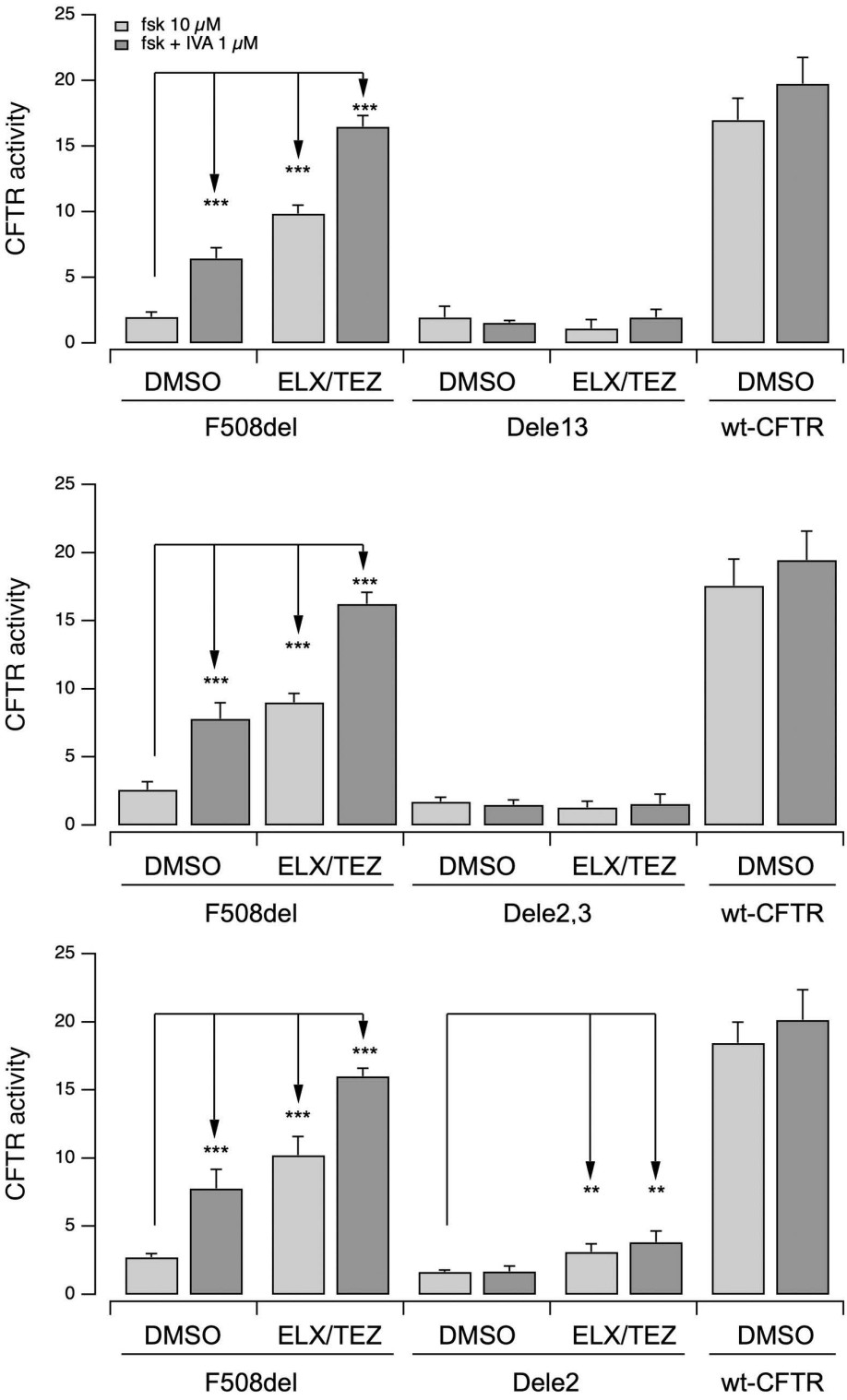

**Fig 4. Functional evaluation of the CFTR activity of the Dele2-, Dele2,3- and Dele13-CFTR mutants on a heterologous expression system after treatment with pharmacological modulators.** The bar graphs show the evaluation of Dele13-(upper panel), Dele 2,3-(middle panel) and Dele2-(lower panel) CFTR activity and for comparison, of F508del- and wildtype-CFTR transiently expressed in CFBE41o- cells stably expressing the HS-YFP. CFTR activity was determined based on the rate of YFP quenching following iodide influx induced by forskolin (fsk; 20 μM) or fsk + iva (1 μM) upon 24h

treatment with vehicle alone (DMSO), or ELX/TZ (3 µM/10 µM). Data were normalized for the activity observed in cells treated with DMSO and stimulated with fsk. Data are shown as means ± SD ($n$ = 3). Asterisks indicate statistical significance of treatments: **, $p < 0.01$; ***, $p < 0.001$.

altered yet full-length CFTR proteins. Moreover, concerning specifically FI077, the mRNA transcribed from the second allele, although lacking exon 13, is in frame.

We thus wondered whether these proteins retain any channel activity or can be somehow rescued upon treatment with modulators.

To this aim, we expressed the in-frame variant proteins, corresponding to the deletion of *CFTR* exon 2 (Dele2; p.(Ser18_Glu54del)) and exon 13 (Dele13; p.Arg560_Glu588del), in CFBE41o-cells cell model with stable expression of the halide-sensitive Yellow Fluorescent Protein (HS-YFP). As controls, we also expressed the protein resulting from the deletion of exons 2 + 3 (Dele2,3; p.Ser18Argfs*16), as well as F508del and wildtype CFTR proteins (Fig 4).

We used the HS-YFP functional assay to quantify CFTR activity. We recorded cell fluorescence before and during the addition of an iodide-containing saline solution. We estimated the HS-YFP quenching rate, that is proportional to CFTR-dependent iodide influx, to assess the rescue of mutant CFTR by pharmacological agents [28]. Prior to the assay, cells were treated for 24 hours with vehicle alone (DMSO) or the indicated correctors (Fig 4). At the time of the assay, cells were acutely stimulated with forskolin (to activate adenylate cyclase and increase intracellular cAMP level) in the absence or presence of the potentiator ivacaftor (Fig 4). Our results showed that the activity of F508del-CFTR improved following treatment with the corrector combination ELX/TZ plus the potentiator ivacaftor (approx. 70–80% of the activity of wild-type CFTR), consistently with previous findings of our and other groups [12,13,32]. On the contrary, Dele13- and Dele2,3- CFTR mutant proteins showed negligible activity, without any rescue following treatment with CFTR modulators. Interestingly, Dele2-CFTR mutant protein displayed a very low transport activity, that was however significantly, although overall modestly, rescued by ELX/TEZ pre-incubation.

This intriguing result prompted us to investigate whether a similar rescue occurs also in a native context. To this purpose, we exploited the patient-derived nasal epithelia, that were treated for 24h with vehicle alone (DMSO) or different CFTR correctors and then, the activity of the CFTR channel was evaluated by means of short-circuit current measurements in Ussing chamber. During the recordings, we sequentially applied different pharmacological agents to dissect the contribution of specific channels and transporters to the transepithelial ion current recorded. First, we blocked the epithelial sodium channel (ENaC) by adding amiloride (10 µM), then we mimicked the increase in intracellular cAMP content (the physiological stimulus that leads to CFTR activation) by adding the membrane-permeable cAMP-analogue CPT-cAMP (100 µM). The potentiator ivacaftor (1 µM) was then added to overcome a possible gating defect and maximize CFTR channel activity. Finally, CFTR-mediated current was blocked by using the CFTR inhibitor-172 (inh-172; 20 µM; [33]) (Fig 5). After the addition of inh-172, the observed drop in the current value was used as an estimate of the total CFTR activity for each epithelium analyzed. Nasal epithelia derived from non-CF donors showed a marked response to the CFTR-stimulating cocktail, with a total CFTR-mediated current that was approximately 25–30 µA/cm$^2$ (Fig 5A). On the contrary, nasal epithelia derived from a CF individual, compound heterozygous for the F508del and 2183AA > G variants, displayed negligible CFTR activity upon treatment with vehicle alone (DMSO) (Fig 5B). However, after 24h treatment with Elexacaftor/Tezacaftor (ELX/TEZ) combination and acute stimulation with CPT-cAMP followed by ivacaftor, the CFTR-mediated current was markedly increased (corresponding to approx. 40% of the activity observed in epithelia derived from non-CF subjects) (Fig 5B). In nasal epithelia derived from donor FI077, treated with vehicle alone (0,18% DMSO), CFTR activity was almost absent and was not increased by acute stimulation with ivacaftor (1.4 ± 0.1 µA; Fig 5C). Interestingly, in epithelia treated for 24h with the ELX/TEZ combination, we observed a modest yet statistically significant rescue of CFTR activity (2.2 ± 0.2 µA; Fig 5C), which is not observed after treatment with Tezacaftor (TEZ) alone (1.6 ± 0.3 µA; Fig 5C).

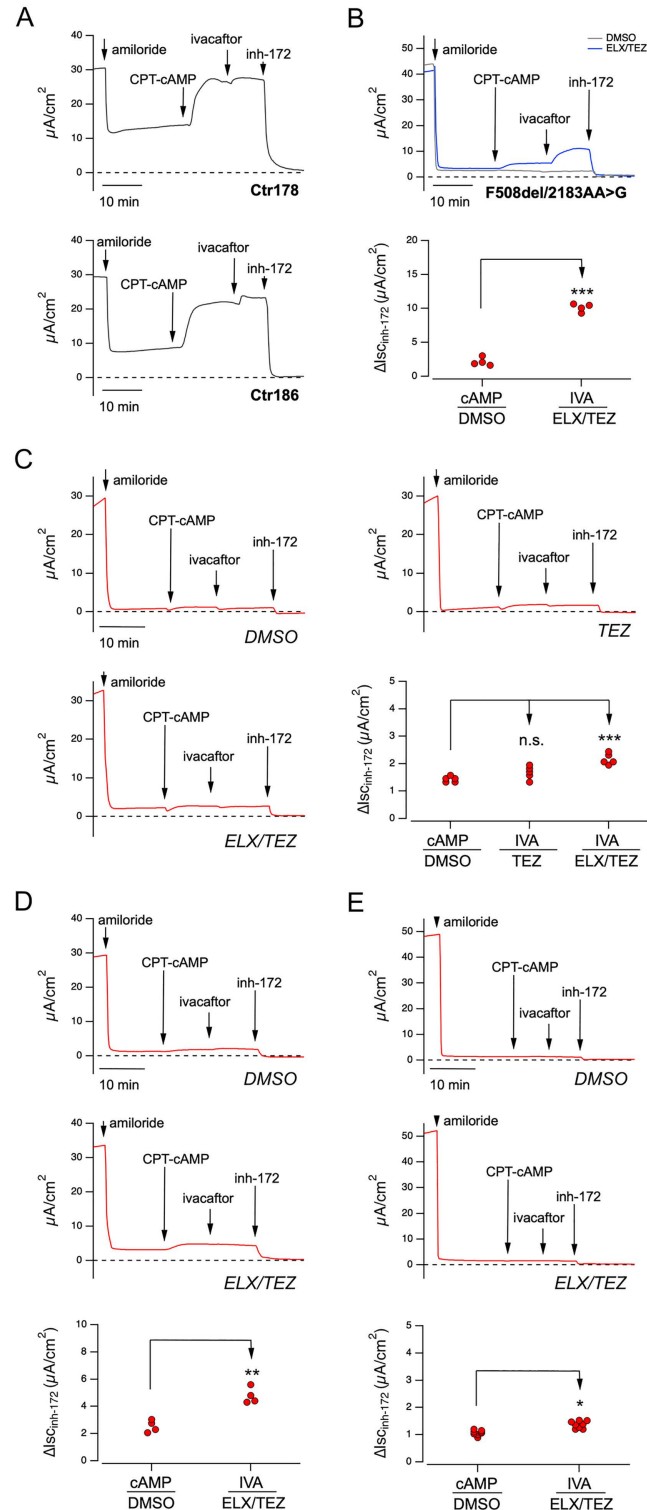

**Fig 5. Functional evaluation of modulators treatment on differentiated nasal epithelia derived from non-CF subjects or from patients with different genotypes.** A A. Representative traces on nasal epithelia derived from two non-CF donors with the short-circuit current technique. During the experiment, the epithelia were treated with amiloride on the apical side (10 µM), CPT-cAMP (100 µM) both apical and basolateral side, ivacaftor (1 µM) and the inhibitor inh-172 (20 µM) on apical side. The dotted line marks the zero current level. B. Representative traces and scatter dot plot summarizing the results of the effect of vehicle alone (DMSO; gray trace) and ELX/TEZ (3 µM/10 µM; blue trace) obtained on nasal epithelia derived from an

individual compound heterozygous for the F508del and 2183AA>G variants, with the short-circuit current technique. During the experiment, the epithelia were sequentially treated as indicated in A. Data reported in the scatter dot plot are the amplitude of the current blocked by 20 µM inh-172 ($\Delta Isc_{inh-172}$). For each experimental condition the number of biological replicates was n = 4. C. Representative traces of the effect of the vehicle alone (DMSO), or Tezacaftor (TEZ, 10 µM), or Elexacaftor/Tezacaftor (ELX/TEZ 3µM/10µM) on nasal epithelia derived from patient FI077 with the short-circuit current technique. Epithelia were acutely stimulated under the same conditions described in A. The scatter dot plot shows the summary of results. Data reported are the amplitude of the current blocked by 20 µM inh-172 ($\Delta Isc_{inh-172}$). For each experimental condition the number of biological replicates was $n$= 6-8. D. Representative traces and scatter dot plot summarizing the results of the effect of short-circuit current measurements performed as in B,C on nasal epithelia derived from MI253 epithelia. For each experimental condition the number of biological replicates was n = 4. E. Representative traces and scatter dot plot summarizing the results of the effect of short-circuit current measurements performed as in B,C on nasal epithelia from FI066 patient. In each experimental condition the number of biological replicates was n = 8. Asterisks indicate statistical significance of treatments: ***, p < 0.001.

To confirm our observations, we recruited one additional patient (donor ID: MI253) bearing the CFTRdele2ins182 complex genomic rearrangement *in trans* with c.870–1110_1113delTAAG. This latter is a deep-intronic mutation promoting the alternative inclusion of a pseudoexon between exon 7 and 8, thus causing a frameshift and the generation of an early stop codon, although a proportion of wild-type mRNA processing may still occur [34]. Nasal epithelia derived from MI253 showed, under control condition, severely reduced CFTR-mediated current ($2.5 \pm 0.4$ µA; Fig 5D), that following ETI treatment increased significantly ($4.8 \pm 0.6$ µA; Fig 5D).

We also analyzed transepithelial chloride secretion in nasal epithelia derived from FI066, carrying the CFTRdele2ins182 and 1717-1G>A variants. In this case, DMSO-treated epithelia exhibited markedly reduced CFTR activity ($1.1 \pm 0.1$ µA; Fig 5E), with a minimal increase following ETI ($1.4 \pm 0.1$ µA; Fig 5E), that, although statistically significant, is of uncertain biological relevance.

The variability in the rescue operated by modulators in the different epithelia cannot be ascribed to differences in the amount of the CFTR transcripts available for translation as we did not find significant differences in the overall abundance of *CFTR* mRNA as assessed by qPCR, nor in the specific expression of the different alleles (S2 Fig).

## Conclusions

Cystic fibrosis is a classic example of how precision medicine can revolutionize the treatment of genetic diseases. The understanding of the functional effect of the different classes of CF-causing mutations has allowed the development of targeted therapies that are improving the lives of many individuals with CF having specific genotypes. However, a significant number of patients still await therapy since they carry rare variants poorly or at all characterized from a functional point of view or expected to function as null alleles and considered not rescuable by definition. Our work underlines the importance of a deep molecular and functional characterization of *CFTR* variants and the relevance of combining functional studies in heterologous systems, suitable to address specific questions, to robust models derived from patients (epithelia differentiated from HNEC), that allow the evaluation in the patient's specific cellular and molecular background. Very interestingly we provide evidence that the well-known CF causing rearrangement CFTRdele2ins182, considered as a null-allele, may be transcribed in alternative forms of *CFTR* mRNA, of which one may lead to the synthesis of a mutant CFTR protein retaining some activity. This may be the starting point to develop and improve novel pharmacological agents able to maintain the expression of such forms while correcting the defective activity.

## Supporting information

**S1 Table. Conversion between Legacy names and HGVS nomenclature of the cited *CFTR* variants.** (PDF)

**S1 Fig. The CFTRdele2ins182 variant induces an alternative splicing of exon 2 and 2/3 in primary HNEC from three unrelated CF patients carrying the variant.** While the molecular characterization of the patient FI077 was in progress, three additional CF patients (donor IDs: FI066, GE202 and MI253) carrying the same complex genomic

rearrangement CFTRdele2ins182 were recruited. The amplification and the sequencing of the RT-PCR products spanning the CFTRdele2ins182 showed, also in these patients, the same pattern of transcripts, with mRNAs lacking exon 2 and both exon 2 and 3, as schematically represented in **A** and confirmed by Sanger sequencing of the RT-PCR products form patient's cDNA spanning *CFTR* exons 1–5 (**B**).
(PDF)

**S2 Fig. Quantitative analysis of *CFTR* transcripts in nasal epithelial cells derived from FI077, FI066 and MI253 individuals by qPCR. A.** Scatter plot showing the overall abundance of *CFTR* transcripts in CF and control individuals measured by qPCR and normalized to β2microglobulin and HPRT housekeeping genes expressed as $2^{-\Delta Ct}$. Each dot represents the mean *CFTR* transcripts overall abundance for each individual. For each subject, two biological replicates (n = 2) were analysed due to the limited availability of samples. **B**. Stacked column chart representing the relative expression of the different transcripts lacking exon 2 or exons 2 and 3 expressed as a percentage of the overall CFTR mRNA set as 100% for each CF sample. The analysis has been performed by qPCR on the RNA obtained from the differentiated epithelia of the three indicated CF samples, with oligonucleotide mapping in the different regions of the CFTR cDNA to quantify the overall and allele-specific abundance of the different transcripts.
(PDF)

**S1 Raw images. Uncropped agarose gel image used for the panel B of Fig 2: Minigene approach for the functional characterization of the 186-13C > G variant.**
(PDF)

## Acknowledgments

We gratefully thank people with CF and their families for their continuous support to our work. We also express gratitude to "Delegazione FFC Ricerca di Genova "Mamme per la ricerca", "Delegazione FFC Ricerca di Tradate Gallarate", "Delegazione FFC di Genova" con "Gruppo di sostegno FFC di Savona Spotorno", "Delegazione FFC Ricerca di Valle Scrivia Alessandria", "Delegazione FFC Ricerca di Montescaglioso", "Delegazione FFC Ricerca di Ascoli Piceno", "Delegazione FCC Ricerca Altomilanese", "Gruppo di sostegno FFC Ricerca di Campiglione Fenile", and "Delegazione FFC Ricerca di Napoli".

## Author contributions

**Conceptualization:** Cristina Pastorino, Nicoletta Pedemonte, Renata Bocciardi.

**Data curation:** Cristina Pastorino, Ludovica Menta, Emanuela Pesce, Valeria Tomati, Valeria Capurro, Serena Cappato, Nicoletta Pedemonte, Renata Bocciardi.

**Formal analysis:** Cristina Pastorino, Ludovica Menta, Emanuela Pesce, Mariateresa Lena, Valeria Tomati, Valeria Capurro, Serena Cappato, Nicoletta Pedemonte, Renata Bocciardi.

**Funding acquisition:** Nicoletta Pedemonte, Renata Bocciardi.

**Investigation:** Cristina Pastorino, Ludovica Menta, Emanuela Pesce, Mariateresa Lena, Valeria Tomati, Valeria Capurro, Serena Cappato.

**Methodology:** Cristina Pastorino, Ludovica Menta, Emanuela Pesce, Mariateresa Lena, Valeria Tomati, Valeria Capurro, Marco Di Duca, Serena Cappato.

**Project administration:** Nicoletta Pedemonte, Renata Bocciardi.

**Resources:** Federico Zara, Nicoletta Pedemonte, Renata Bocciardi.

**Software:** Nicoletta Pedemonte, Renata Bocciardi.

**Supervision:** Vito Terlizzi, Andrea Gramegna, Francesco Blasi, Carlo Castellani, Federico Zara, Nicoletta Pedemonte, Renata Bocciardi.

**Validation:** Nicoletta Pedemonte, Renata Bocciardi.

**Visualization:** Cristina Pastorino, Ludovica Menta, Nicoletta Pedemonte, Renata Bocciardi.

**Writing – original draft:** Cristina Pastorino, Nicoletta Pedemonte, Renata Bocciardi.

**Writing – review & editing:** Vito Terlizzi, Andrea Gramegna, Francesco Blasi, Carlo Castellani, Federico Zara, Nicoletta Pedemonte, Renata Bocciardi.

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
