## [Decision Letter · Decision Letter 0]

29 Jul 2025

PONE-D-25-36949Alternative splicing and residual function potentially expand the therapeutic landscape of the CFTRdele2ins182 variant.PLOS ONE

Dear Dr. Pedemonte,

Thank you for submitting your manuscript to PLOS ONE. After careful consideration, we feel that it has merit but does not fully meet PLOS ONE’s publication criteria as it currently stands. Therefore, we invite you to submit a revised version of the manuscript that addresses the points raised during the review process.

We look forward to receiving your revised manuscript.

Kind regards,

Yoshi

Prof. Yoshinori Marunaka, MD. PhD

Academic Editor

PLOS ONE

Journal Requirements:

“Fondazione per la Ricerca sulla Fibrosi Cistica grants n. FFC #3/2023, n.FFC #9/2019 and n. FFC #10/2021; Italian Ministry of Health [grant n. GR-2018–12367126, PNRR-MR1-2023-12378412] and through Cinque per mille (5Xmille 2019 grant 5M-2019-23680413) and Ricerca Corrente.”

5. Please ensure that you refer to Figure 4 in your text as, if accepted, production will need this reference to link the reader to the figure.

Reviewers' comments:

Reviewer's Responses to Questions

**Comments to the Author**

1. Is the manuscript technically sound, and do the data support the conclusions?

Reviewer #1: Yes

Reviewer #2: Yes

2. Has the statistical analysis been performed appropriately and rigorously? 

Reviewer #1: No

Reviewer #2: Yes

3. Have the authors made all data underlying the findings in their manuscript fully available?

Reviewer #1: No

Reviewer #2: Yes

4. Is the manuscript presented in an intelligible fashion and written in standard English?

Reviewer #1: Yes

Reviewer #2: Yes

5. Review Comments to the Author

Reviewer #1: This manuscript presents an analysis of rare CFTR mutations, with a particular focus on the complex allele [186-13C>G; 1898+3A>G] and the CFTRdele2ins182 variant, both of which have not been extensively characterized to date. The findings provide valuable insights into the CFTR mutation spectrum and their functional consequences. While the data overall support the conclusions, several revisions are necessary prior to publication:

Figure quality: The resolution of the figures is generally low and should be improved to ensure data clarity and readability.

Figure 5: If available, time-course data should be included to better illustrate the dynamics of the observed effects. Furthermore, appropriate statistical analyses should be conducted and clearly presented, as no statistical tests appear to have been applied.

Figure 6E: Only a scatter plot of the data is shown. Representative traces from the corresponding experiments should also be provided to support the quantitative data.

Short-circuit current recording (CF-HNE): The transepithelial resistance (TER) values should be reported to validate the quality and consistency of the epithelial monolayers used.

Methods – HS-YFP: The authors state that halide-sensitive YFP was used, but the specific YFP variant is not clearly described. Detailed information on the YFP construct (e.g., mutation sites or reference to the original plasmid) should be included for reproducibility.

Reviewer #2: This is an interesting paper manuscript, detailing that the CFTRdele2ins182 variant, previously considered a null allele, is capable of producing a defective CFTR protein, potentially responsive to modulator rescue therapy.

The laboratory work is detailed and well explained.

6. PLOS authors have the option to publish the peer review history of their article (what does this mean? ). If published, this will include your full peer review and any attached files.

**Do you want your identity to be public for this peer review?** For information about this choice, including consent withdrawal, please see our Privacy Policy .

Reviewer #1: **Yes: ** Tsukasa Okiyoneda

Reviewer #2: No

---

## [Author Response · Author response to Decision Letter 1]

7 Aug 2025

Editorial comments

Reply: We have modified manuscript formatting and file naming to meet PLOS ONE's style requirements

“Fondazione per la Ricerca sulla Fibrosi Cistica grants n. FFC #3/2023, n.FFC #9/2019 and n. FFC #10/2021; Italian Ministry of Health [grant n. GR-2018–12367126, PNRR-MR1-2023-12378412] and through Cinque per mille (5Xmille 2019 grant 5M-2019-23680413) and Ricerca Corrente.”

Reply: The requested statement has been added, and the change has been included in the Cover letter of the revised manuscript

Reply: The document including the uncropped gel image was already provided at the first submission. However, thanks to the editorial comment we realized that the provided uncropped image was not properly annotated. We have thus resolved this issue by providing the document with the requested details (S1_raw_image_REV).

Reply: Following this comment, we are now presenting the relevant data as supplementary Figure S2.

5. Please ensure that you refer to Figure 4 in your text as, if accepted, production will need this reference to link the reader to the figure.

Reply: The legend to Figure 4 (page 16) has been maintained in the main text by mistake, as the corresponding figure has been moved to the Supplementary material as Figure S1. The numbering of Figures included in the main text is thus changed from 1 to 5. All the Figures are properly cited in the text.

Reply: No request to include citation of previous article has been raised by Reviewers.

Reply: the list of references has been checked and is complete and correct, we did not cite any retracted articles.

Review Comments to the Author

Reviewer #1: This manuscript presents an analysis of rare CFTR mutations, with a particular focus on the complex allele [186-13C>G; 1898+3A>G] and the CFTRdele2ins182 variant, both of which have not been extensively characterized to date. The findings provide valuable insights into the CFTR mutation spectrum and their functional consequences.

Reply: We thank the reviewer for the positive comment.

While the data overall support the conclusions, several revisions are necessary prior to publication:

Figure quality: The resolution of the figures is generally low and should be improved to ensure data clarity and readability.

Reply: We apologize for the low quality of the figures. However, this was caused by the conversion of the high-quality tiff images (provided by us) into pdf during the uploading of the files to generate the single document used for reviewing purposes. The high-quality figures will be eventually used for publication if the manuscript is accepted.

Figure 5: If available, time-course data should be included to better illustrate the dynamics of the observed effects. Furthermore, appropriate statistical analyses should be conducted and clearly presented, as no statistical tests appear to have been applied.

Reply: We could not perform a time-course analysis of drug efficacy due to constraints intrinsic to the YFP-based functional assay and to the very low activities measured for this CFTR variants. However, the time points chosen for the experiments are the most robust and commonly used (48 h post-transfection, 24 h drug treatment) for this type of analysis. As requested by the reviewer, we are now including in the figure the statistical analysis of the data.

Figure 6E: Only a scatter plot of the data is shown. Representative traces from the corresponding experiments should also be provided to support the quantitative data.

Reply: According to reviewer’s comment, we have modified the figure (now revised Fig. 5) to include also representative traces recorded for patient FI066.

Short-circuit current recording (CF-HNE): The transepithelial resistance (TER) values should be reported to validate the quality and consistency of the epithelial monolayers used.

Reply: As requested, we are now reporting, in the Material and Methods section, the data on the transepithelial resistance (TER) values to validate the quality and consistency of the epithelial monolayers used.

Methods – HS-YFP: The authors state that halide-sensitive YFP was used, but the specific YFP variant is not clearly described. Detailed information on the YFP construct (e.g., mutation sites or reference to the original plasmid) should be included for reproducibility.

Reply: We have added this information in the Material and Methods, Cell Culture section, including the citation to the original report that describes the YFP variant and provides information on the plasmid.

Reviewer #2: This is an interesting paper manuscript, detailing that the CFTRdele2ins182 variant, previously considered a null allele, is capable of producing a defective CFTR protein, potentially responsive to modulator rescue therapy.

The laboratory work is detailed and well explained.

Reply: We thank the reviewer for the generous praise.

---

## [Decision Letter · Decision Letter 1]

10 Aug 2025

Alternative splicing and residual function potentially expand the therapeutic landscape of the CFTRdele2ins182 variant.

PONE-D-25-36949R1

Dear Dr. Pedemonte,

We’re pleased to inform you that your manuscript has been judged scientifically suitable for publication and will be formally accepted for publication once it meets all outstanding technical requirements.

Kind regards,

Prof. Yoshinori Marunaka, MD. PhD

Academic Editor

PLOS ONE

Additional Editor Comments (optional):

Reviewers' comments:

Reviewer's Responses to Questions

**Comments to the Author**

1. If the authors have adequately addressed your comments raised in a previous round of review and you feel that this manuscript is now acceptable for publication, you may indicate that here to bypass the “Comments to the Author” section, enter your conflict of interest statement in the “Confidential to Editor” section, and submit your "Accept" recommendation.

Reviewer #1: All comments have been addressed

2. Is the manuscript technically sound, and do the data support the conclusions?

Reviewer #1: Yes

3. Has the statistical analysis been performed appropriately and rigorously? 

Reviewer #1: Yes

4. Have the authors made all data underlying the findings in their manuscript fully available?

Reviewer #1: Yes

5. Is the manuscript presented in an intelligible fashion and written in standard English?

Reviewer #1: Yes

6. Review Comments to the Author

Reviewer #1: The authors have addressed the majority of the concerns I raised, and the revisions appear satisfactory overall. I consider this paper acceptable for publication.

7. PLOS authors have the option to publish the peer review history of their article (what does this mean? ). If published, this will include your full peer review and any attached files.

**Do you want your identity to be public for this peer review?** For information about this choice, including consent withdrawal, please see our Privacy Policy .

Reviewer #1: **Yes: ** Tsukasa Okiyoneda

---

## [Editor Report · Acceptance letter]

PONE-D-25-36949R1

PLOS ONE

Dear Dr. Pedemonte,

I'm pleased to inform you that your manuscript has been deemed suitable for publication in PLOS ONE. Congratulations! Your manuscript is now being handed over to our production team.

Kind regards,

on behalf of

Professor Yoshinori Marunaka

Academic Editor

PLOS ONE